# Oncolytic Adenoviruses: The Cold War against Cancer Finally Turns Hot

**DOI:** 10.3390/cancers14194701

**Published:** 2022-09-27

**Authors:** Bryan Oronsky, Brian Gastman, Anthony P. Conley, Christopher Reid, Scott Caroen, Tony Reid

**Affiliations:** 1EpicentRx, Torrey Pines, La Jolla, CA 92037, USA; 2Cleveland Clinic, Cleveland, OH 44195, USA; 3University of Texas MD Anderson Cancer Center, Houston, TX 77030, USA

**Keywords:** oncolytic adenoviruses, checkpoint inhibitors, cancer immunity cycle, immunosuppression, priming

## Abstract

**Simple Summary:**

Immunotherapy has revolutionized cancer treatment, as demonstrated by the tremendous success of checkpoint inhibitors in different tumor types. Unfortunately, most patients, particularly patients with non-responsive “cold” tumors, do not benefit from checkpoint inhibitors. Enter “armed” oncolytic viruses, which “cooperate” with checkpoint inhibitors to improve anticancer responses. These are genetically engineered viruses that selectively infect, replicate in, and kill cancer cells but not cells from healthy tissues; in the process, oncolytic viruses express the therapeutic proteins that they are armed with or carry. This effectively turns the infected tumors “hot” and makes them suitable for treatment with checkpoint inhibitors. The most well-studied of all the oncolytic viruses are adenoviruses. These are agents of the common cold, which makes them remarkably safe for clinical use. This review article summarizes the oncolytic adenoviruses in advanced clinical trials and presents strategies to improve their anticancer activity.

**Abstract:**

Oncolytic viruses, colloquially referred to as “living drugs”, amplify themselves and the therapeutic transgenes that they carry to stimulate an immune response both locally and systemically. Remarkable exceptions aside, such as the recent 14-patient trial with the PD-1 inhibitor, dostarlimab, in mismatch repair (MMR) deficient rectal cancer, where the complete response rate was 100%, checkpoint inhibitors are not cure-alls, which suggests the need for a combination partner like oncolytic viruses to prime and augment their activity. This review focuses on adenoviruses, the most clinically investigated of all the oncolytic viruses. It covers specific design features of clinical adenoviral candidates and highlights their potential both alone and in combination with checkpoint inhibitors in clinical trials to turn immunologically “cold” and unresponsive tumors into “hotter” and more responsive ones through a domino effect. Finally, a “mix-and-match” combination of therapies based on the paradigm of the cancer-immunity cycle is proposed to augment the immune responses of oncolytic adenoviruses.

## 1. Introduction

A major challenge to successful anticancer treatment, especially with immune checkpoint inhibitors (ICBs) specific for CTLA-4, PD-1 and PD-L1 and chimeric antigen T (CAR-T) cell therapy, is the presence of immunologically “cold” or non-T-cell inflamed tumors, which have prompted countermeasures to heat them up vis-à-vis T cell infiltration [1].

One of these countermeasures is oncolytic viruses (OVs), colloquially referred to as “living drugs” [2]. Several oncolytic viruses have received regulatory approval. These include Talimogene laherparepvec (T-VEC), an attenuated herpes simplex virus, type 1 (HSV-1) for melanoma, Delytact (teserpaturev), another HSV-1 virus, in Japan for the treatment of glioma [3], and Rigavir, an unmodified ECHO-7 virus in Latvia, Georgia, and Armenia and H101, an oncolytic adenovirus approved in China [4]. To date, however, the full benefit of combination with checkpoint inhibitors has not been realized in large phase 3 trials, which argues for improvements in the design of OVs. Other OVs have entered clinical trials, including poxviruses, HSV-1, coxsackieviruses, poliovirus, measles virus, Newcastle disease virus (NDV), and reovirus.

The most studied and widely used oncolytic viruses are adenoviruses. These are non-enveloped double-stranded DNA viruses with linear genomes of ~30–38 kb and a fiber-covered icosahedral protein capsid [5,6]. Most genetically engineered oncolytic adenoviruses are derived from Ad serotype 5 (Ad5) and Ad serotype 2 (Ad2), while over 100 different antigenic serotypes and 7 different species (A–G) have been identified, which infect mammals (genus mastadenoviruses) and birds (genus aviadenoviruses) [7]. The adenoviral replication cycle is broadly divided into two temporal phases with early (E1A, E1B, E2A, E2B, E3, and E4) and late (L1–L5) transcription units, as shown in the figure below; the former is responsible for DNA synthesis and the latter for the structural proteins of the Ad virion [8]. The Ad genome is flanked by inverted terminal repeat (ITR) sequences, which initiate replication (Figure 1).

H101, an E1B-55K gene deleted recombinant Ad5 and the successor to ONYX-015, which was the first tumor-specific oncolytic adenovirus (OAV) evaluated in the clinic, received approval from the Chinese FDA for the treatment of nasopharyngeal carcinoma in combination with cisplatin and/or 5-fluorouracil (5-FU) [9,10,11]. Since ONYX-015 and H101, several “generations” of conditionally replicative adenoviruses have followed, which include diverse modifications to E1A and E1B and the capsid, the deletion or partial deletion of E3, and the insertion of therapeutic transgenes.

Cancer is a systemic disease, such that by the time tumors “go live”, that is, reach 1–2 mm in diameter and acquire a vasculature, circulating cancer cells are present [12]. Nevertheless, total local surgical resection is curative in most patients, despite the ab initio systematicity of cancer cells. This suggests that an immune response is involved, possibly from the release of potential tumor antigens, pro-inflammatory cytokines and chemokines, and other danger signals during surgery [13].

Similarly, oncolytic adenoviruses (OAVs) can elicit and redirect both innate and adaptive immune responses to target tumors. This is accomplished through selective infection, replication, and direct elimination of cancer cells, including cancer stem cells, which contribute to therapeutic resistance and recurrence, the release of danger signals and tumor-associated antigens (TAAs) and tumor-specific antigens (TSAs), as well as expression of transgene-encoded immunomodulatory proteins [14].

One of the main limitations to the success of these OAVs as chemo- and immune-sensitizers is the degree of immunosuppression present in the tumor microenvironment (TME). Multiple mechanisms are responsible for the maintenance of this immunosuppressive phenotype. These mechanisms include upregulation of procancerous factors such as IL-10, IL-18, VEGF, Prostaglandin E, and TGF-β, infiltration of regulatory T cells (T_reg_ cells), and myeloid-derived suppressor cells (MDSCs), increased deposition of extracellular matrix or fibrosis, and overexpression of checkpoint ligands, such as programmed cell death ligand 1 (PDL1) or its cognate receptor, PD-1, cytotoxic T-lymphocyte protein 4 (CTLA4), TIM-3 (HAVcr2), LAG-3 (CD223), TIGIT, B7-H3 (CD276), B7-H4 (VCTN1), downregulation or loss of HLA class I molecules, and A2aR and decreased neoepitope availability [15,16].

The effectiveness of oncolytic adenovirotherapy to bring about cold to hot transformation in the TME critically depends on at least 3 factors: (1) degree of attenuation since the more modifications which are made to the viral genome to increase safety or to ablate native tropism and improve tumor targetability, for example, the more the potency of the virus and its ability to induce a sufficient antitumor response are compromised; (2) choice of immunomodulatory payload since, for example, granulocyte-macrophage colony-stimulating factor (GM-CSF), by far the most widely used transgene in OAVs, may contribute to tumor growth and immunosuppression [17]. In addition, IL-2, IL-12, and Tumor Necrosis Factor Alpha (TNF-a), all well-studied immunostimulatory transgenes, are also associated with immunosuppression (and, interestingly, intralesional mRNA injections of these factors have not performed so well); (3) presence of the coxsackievirus and adenovirus receptor (CAR), which mediates viral attachment and infection [18,19,20].

This review covers specific design features of clinical adenoviral candidates and highlights their potential both alone and in combination with checkpoint inhibitors in clinical trials that were recently completed or are currently active to turn immunologically “cold” and unresponsive tumors into “hotter” and more responsive ones.

## 2. Adenoviral Clinical Candidates

Adenovirus is arguably the vector of choice for the treatment of cancer because of its long safety track record both in and out of oncology since the early 1990s, low pathogenicity, lack of host genomic integration, relative ease of manufacture, strong immunogenicity, and capacity for transgene incorporation [21]. A search as of August 2022 on clinical trials.gov found seven ongoing or recently completed early-stage clinical trials with conditionally replicating oncolytic adenoviruses (OAVs) for which interim or final data is known or has been reported. The clinical status and design features of these OAVs, which are presented individually below, include AdAPT-001 and AdAPT-039, CG0070, Enadenotucirev, NG-350A and NG-641, ONCOS-102, LOAd703, VCN-01, and OBP-301.

## 3. AdAPT-001 and AdAPT-039 (EpicentRx)

AdAPT-001 is the first conditionally replicating adenovirus to be presented out of familiarity because, full transparency, coauthors Tony Reid and Chris Larson not only conceived and designed AdAPT-001 but also led its clinical development and, more importantly, for this review, because AdAPT-001 presents some distinctive design characteristics, which are instructive to compare and contrast with the other OAVs that follow. Firstly, the AdAPT-001 Ad5 base oncolytic vector is not targeted to tumors through capsid modification, the insertion of exogenous cancer-specific promoters, or hybridization of adenoviral serotypes, strategies that are currently in clinical use to modify the natural tropism of adenoviruses. Rather, AdAPT-001 is detargeted from non-tumor cells through the deletion of a small 50 base pair region located upstream of the E1A initiation site, which contains multiple transcription factor binding sites that are indispensable in non-tumor cells, leading to abortive infection and no or restricted cytolytic activity, but dispensable in tumors, where potent near wild type levels of replication, expression, and cytolytic activity are observed [22,23,24].

The premise behind such a minimal modification is that too many viral additions or deletions may significantly change viral biological features and activity. Due to this “less is more” emphasis, not only is the cytolytic efficiency of AdAPT-001 comparable to that of wild-type virus, but so are the yields during manufacture. In fact, AdAPT-001 is manufacturable to cGMP standards “in house”, which removes reliance on often inefficient and costly contract manufacturing organizations (CMO).

The other modification in AdAPT-001 is the deletion of the E1B19K gene—a Bcl-2 adenoviral homolog that potently inhibits apoptosis [25]—and its replacement with a Transforming Growth Factor-beta (TGF-β) ligand “trap”. This trap is a TGFβ receptor ectodomain-IgG Fc fusion protein, which binds to and neutralizes the immunosuppressive and fibrosis-inducing cytokine, TGF-β [26].

Administered by intratumoral (IT) injection every 2 weeks at a dose of 1 × 10^12^ vps, AdAPT-001 is currently in Phase I/II study called BETA PRIME (NCT04673942) for patients with treatment-refractory, metastatic cancers both as monotherapy in Part 1, which is almost complete, and in combination with a checkpoint inhibitor in Part 2, which has not yet started. Preliminary data demonstrate that AdAPT-001 is not only well-tolerated but also active in TGFβ-driven tumors.

AdAPT-039 is a folate-targeted nanoparticle formulation of AdAPT-001 to bypass not only pre-existing neutralizing immunity but also the CAR receptor dependency of Ad cell entry. A Phase I trial with AdAPT-039 is scheduled to start shortly.

## 4. CG0070 (CG Oncology)

CG0070 is a conditionally replicating type 5 adenovirus that selectively replicates in retinoblastoma (Rb) pathway-defective bladder tumor cells. This highly modified virus carries the cancer-selective promoter E2F-1 in place of the wild-type adenovirus E1A promoter and the cytokine granulocyte macrophage colony stimulatory factor (GM-CSF) in place of the deleted E3 region [27]. CG0070 is only used in well confined intravesical bacillus Calmette-Guérin (BCG)-resistant non-muscle invasive bladder cancer (NMIBC), a highly curable tumor type that may nevertheless progress to muscle-invasive disease in the absence of effective treatment [28,29].

In a phase II trial of 66 BCG-unresponsive NMIBC patients that received intravesical CG0070, the 6-month CR was 47% (95% CI 32–62%), 58% in the carcinoma in situ (CIS) group, and 33% in the Ta/T1 group. Treatment was well tolerated. CG0070 is currently under investigation for 110 patients with BCG-unresponsive NMIBC as monotherapy in the phase III registration trial (BOND-003, NCT04452591). It is administered in a weekly induction course x 6 at a dose of 1 × 10^12^ vps, followed by a second weekly induction course at a dose of 1 × 10^12^ vps for non-responders, and a maintenance course of weekly x 3 at a dose of 1 × 10^12^ vps for completer responders. A phase II trial of CG0070 + pembrolizumab (CORE-001, NCT04387461) is also actively recruiting [30,31].

## 5. Enadenotucirev (EnAd) and NG-350A and NG-641 (Psioxus Therapeutics)

Enadenotucirev (EnAd), formerly ColoAd1, is the product of ‘directed evolution’, having been iteratively pooled and passaged to replicate only in cancer cells and is mostly administered intravenously [32]. EnAd has been investigated in several Phase I clinical trials, which collectively established the safety, tolerability, and pro-immunogenic effects of intravenous and intratumoral administration. These Phase I trials include NCT02028117 (OCTAVE) with EnAd plus paclitaxel in recurrent platinum-resistant ovarian cancer, NCT03916510 in rectal cancer with capecitabine and radiation, and NCT02636036 in solid tumors with the PD-1 inhibitor, nivolumab. However, the only results reported are for OCTAVE, in which enadenotucirev plus paclitaxel demonstrated manageable safety, an encouraging median PFS, and increased tumor immune-cell infiltration [33]. To augment immune responses, two EnAd variants are under investigation in phase I clinical trials: NG-350A (NCT03852511), which expresses a fully agonistic CD40 antibody, and NG-641 (NCT04053283), which quadrivalently expresses the bispecific T-cell engager (BiTE) FAP/CD3, chemokine ligands 9 and 10 (CXCL9 and CXCL10) and interferon alpha (IFNα) [34].

## 6. ONCOS-102 (Targovax)

ONCOS-102 is an oncolytic adenovirus with a fiber shaft and tail domain of HAdV-5 and a fiber knob domain of HAdV-3. The GM-CSF gene replaces the E3 6.7K/gp19K gene. Accordingly, transduction is mediated by the desmoglein 2 receptor instead of the often downregulated or deficient coxsackie and adenovirus receptor (CAR) to which the fiber from Ad5 binds [35]. The virus carries a deletion of 24 base pairs in the retinoblastoma (Rb) binding domain of the E1A region. Because of this deletion, ONCOS-102 selectively targets only those tumors with Rb protein pathway disruption [36]. A human GM-CSF transgene is inserted in the E1B19K gene region.

ONCOS-102 has completed several clinical trials. Of particular interest is NCT030036, a 20-patient pilot study of ONCOS-102 plus the PD-1 inhibitor, pembrolizumab, in PD-1 inhibitor-refractory melanoma, for which FDA fast track designation was awarded based on a 35% objective response rate (ORR), i.e., complete, or partial responses according to the Response Evaluation Criteria in Solid Tumors (RECIST) 1.1. No DLTs were observed, and the most common adverse events were chills and fever from Ad replication [37].

However, a phase I/II study of ONCOS-102 with durvalumab, an anti-PD-L1 antibody, for the treatment of advanced peritoneal malignancies (NCT02963831) did not meet its efficacy endpoints [38].

## 7. LoAd-703 (Lokon Pharma)

This is another heavily modified type 5 oncolytic adenovirus with an Ad35 fiber, a 24 base pair deletion in the retinoblastoma (Rb) binding domain of the E1A region, and a partial deletion in E3 to express two transgenes, both of which are under the control of a cytomegalovirus (CMV) promoter: 1) a trimerized (TMZ) form of the membrane-bound CD40 ligand (CD40L), which binds to CD40, a cell surface molecule on antigen-presenting cells, and 2) the ligand for the signaling domain 4-1BB (4-1BBL), which binds to the costimulatory receptor, 4-1BB. CD40/CD40L and 4-1BB/4-1BBL interactions are critical for the development antigen-specific cytotoxic CD8+ T-cell responses [39,40]. Provided that major histocompatibility complex (MHC) class I is present, the co-expression of CD40L and 4-1BBL on LoAd-703-infected tumor cells may synergistically contribute to reverse T cell anergy with checkpoint inhibitors and chemotherapies.

In Phase I/II unresectable or metastatic pancreatic cancer trial (NCT02705196), 18 evaluable patients, the majority of which were previously treated, received intratumoral injections of LoAd-703 plus intravenously delivered nab-paclitaxel and gemcitabine. The safety profile was manageable. The reported overall response rate (ORR) was 44%, the disease control rate (DCR) was 94%, and the median overall survival (OS) was 8.7 months, which compares favorably with an overall historical survival of 6.8 months in patients that receive nab-paclitaxel and gemcitabine [41]. The proportion of T effector memory cells significantly increased while the proportion of T regulatory cells and myeloid-derived suppressor cells significantly decreased. A follow-up clinical trial (NCT02705196), which combines LOAd-703, nab-paclitaxel, and gemcitabine, and the anti-PDL-1 inhibitor atezolizumab is ongoing [42]. Trials are also ongoing in checkpoint inhibitor refractory malignant melanoma with the PD-L1 inhibitor, atezolizumab, (NCT04123470) and in colorectal cancer in combination with atezolizumab (NCT03555149).

## 8. VCN-01 (Synthetic Biologics, Formerly VCN Biosciences)

VCN-01 is an E1A 24 bp-deleted (for selective replication in Rb deficient tumors), and partially E3 deleted type 5 oncolytic adenovirus that expresses hyaluronidase for degradation of the tumor extracellular matrix (ECM), which is especially prominent in pancreatic adenocarcinoma. To eliminate dependence on CAR binding, the capsid fiber incorporates an arginine glycine aspartate (RGD) integrin-binding motif. In a Phase I trial for patients with pancreatic adenocarcinoma that received intravenously administered VCN-01 nab-paclitaxel plus gemcitabine (NCT02045589), the safety profile was manageable, and viremia was observed as well as increased levels of immune biomarkers [43].

## 9. OBP-301 (Telomelysin) (Oncolys BioPharma)

OBP-301 is an ‘unarmed’ type 5 adenovirus in which the human telomerase reverse transcriptase (hTERT) promoter has been inserted upstream of the E1 genes to drive tumor-specific expression.

In a Phase I dose-escalation study of endoscopic intratumoral injection of 10^10^, 10^11^, and 10^12^ vp of OBP-301 (Telomelysin) with 60 Gy radiotherapy over 6 weeks in 13 esophageal cancer (NCT03213054) patients deemed unfit for standard treatments, the objective response rate was 91.7%, and the complete response rate was 83.3% in stage I and 60.0% in stage II/III concomitant with massive infiltration of CD8+ cells and increased PD-L1 expression [44]. This suggests clinical synergy with a checkpoint inhibitor.

A Phase II trial in combination with pembrolizumab in esophagogastric adenocarcinoma (NCT03921021) is ongoing.

## 10. DNX-2401 (Tasadenoturev) (DNAtrix—A Spin-Off of University of Texas MD Anderson Cancer Center)

DNX-2401 is an E1A 24 bp-deleted type 5 oncolytic adenovirus that selectively replicates in Rb-deficient tumors, and that incorporates an arginine glycine aspartate (RGD) integrin-binding motif, which mediates viral attachment and entry instead of CAR.

In a phase I trial (NCT00805376), 37 patients with recurrent malignant glioma received a single intratumoral injection of DNX-2401 over eight dose levels (group A; *n* = 25) or underwent intratumoral injection through a permanently implanted catheter, followed 14 days later by *en bloc* resection to acquire post-treatment specimens (group B; *n* = 12). In group A, 20% of patients survived > 3 years from treatment, and a ≥ 95% tumor reduction was observed in 3 patients, resulting in > 3 years of progression-free survival. Analyses of post-treatment surgical specimens documented direct virus-induced oncolysis and infiltration of CD8^+^ cells [45].

In another Phase I trial (NCT03178032), 12 newly diagnosed pediatric patients (3–12 years old) with diffuse intrinsic pontine glioma (DIPG), an untreatable and universally fatal brain tumor, received a single infusion of DNX-2401 through a catheter placed in the cerebellar peduncle at doses of 1 × 10^10^ (first four patients) or 5 × 10^10^ (next eight patients) viral particles (vp) followed by subsequent radiotherapy. Activity was demonstrated with a partial response in three patients and stable disease in eight patients. Median progression-free survival was 10.7 months. Median overall survival was 17.8 months. However, four treatment-related Grade 3 neurological adverse events occurred [46]. Based on these results, DNX-2401 has been granted FDA Fast Track and Orphan designation and EMA PRIME and Orphan designation.

In a Phase II study (NCT02798406) where 49 patients with recurrent glioblastoma (GBM) received 200 mg pembrolizumab every three weeks + a single intratumoral injection of DNX-2401, the median overall survival was 12.5 months, which compares favorably with the standard of care agents, lomustine and temozolomide, where the median overall survival is approximately7.2 months. The adverse event profile was manageable. A Phase 3 trial in recurrent GBM is reportedly planned [47].

## 11. Ad5-yCD/mutTKSR39rep-hIL12 (Henry Ford Health System)

Like ONYX-015 and H101, this is an E1B-55k-deleted Ad5 virus with an insertion of yeast cytosine deaminase (yCD), mutant herpes simplex thymidine kinase_SR39_ (TKSR39), and human interleukin (IL)-12, which is under investigation in pancreatic and prostate cancer. yCD and TKSR39 convert the prodrugs 5-fluorocytosine (5-FC) and valganciclovir to their toxic forms in infected cells. In a Phase I trial, 12 patients with metastatic pancreatic cancer (T2N0M1-T4N1M1) received intratumoral injections of Ad5-yCD/mutTKSR39rep-hIL12 at escalating doses (1 × 10^11^, 3 × 10^11^, or 1 × 10^12^ viral particles) in combination with 5-fluorocytosine (5-FC) therapy for 7 days followed by chemotherapy (FOLFIRINOX or gemcitabine/albumin-bound paclitaxel (nab-paclitaxel)). The safety profile was acceptable; no MTD was reached, which is encouraging because systemic administration of IL-12 is associated with severe dose-limiting toxicities [48]. The median overall survival (OS) of the 6 patients that received Ad5-yCD/mutTKSR39rep-hIL12 at a dose of 1 × 10^12^ viral particles was 18.1 months, which exceeds the OS that is historically associated with FOLFIRINOX and gemcitabine + nab-paclitaxel of 11.1 and 6.8 months, respectively, although it is difficult to draw conclusions about efficacy based on such a small sample size [49].

## 12. CELYVIR (Hospital Infantil Universitario Niño Jesús Madrid, Spain)

CELYVIR is an intravenously administered formulation of autologous mesenchymal stem cells (MSCs) that carry ICOVIR-5, a heavily modified Ad5 dependent on an aberrant RB pathway, with a 24 base pair deletion, an RGD insertion, and an E2F-1 promoter insertion that failed to demonstrate activity in a Phase I melanoma trial [50].

In a Phase I pediatric trial (1–18 years) with advanced relapsed/refractory solid tumors, 15 patients, 9 of whom were evaluable, received Celyvir manufactured with MSCs collected from a bone marrow aspirate and then given IV weekly for 6 weeks at doses from 2 × 10^6^ cells/Kg and 2 × 10^4^ viral particles (vp) per cell. The safety profile was tolerable. Stable disease was reported in two patients with neuroblastoma, but no radiologic responses were seen [51].

A table, which summarizes these adenoviral clinical candidates is shown below (Table 1).

## 13. Discussion and Conclusions

The use of immunotherapy to treat cancer is a highly *en vogue* topic with literally thousands of published articles on it and multiple clinical trials underway. Amid all the (understandably) enthusiastic claims about the “game-changing” properties of checkpoint inhibitors (CPIs) and the promise of total tumor eradication, which most recently occurred with the PD-1 inhibitor, dostarlimab, in 14 locally advanced mismatch repair (MMR) deficiency rectal cancer patients [52], success stories are limited to a small subset of the treatment population in an even smaller subset of proimmunogenic and genetically unstable tumors such as NSCLC, melanoma, and MMR deficient rectal cancer [53]. Rarer still are those patients with long-term durable remissions since acquired resistance after initial response to CPIs is the rule, not the exception [54]. Moreover, despite a series of positive trials with anti-PD-1/L1 checkpoint inhibitors, several other immunotherapies such as the anti-TIGIT agent, tiragolumab, the engineered interleukin-2, nemvaleukin alfa, and indoleamine 2,3-dioxygenase 1 inhibitors (IDO1) have failed [55]. Also, the combination of checkpoint inhibitors and targeted therapies, while promising, is still early stage and associated with complex and unexpected toxicities [56]. This is the thorniest challenge in oncology—how, and in combination with what *exactly* (exactly being the operative word), to overcome resistance to checkpoint inhibitors, and immunotherapy, in general, so that all patients durably benefit, regardless of tumor type and pre-existing immunogenicity of tumor type.

To date, no universal solution has presented itself with combinations that include other checkpoint inhibitors, cytokines, cytokine inhibitors, epigenetic inhibitors, adoptive cell transfer, antiangiogenics, bispecific T cell engagers, chemotherapies, radiotherapy, targeted therapies, antitumor vaccines, oncolytic viruses, etc. In this regard, the seminal cancer-immunity cycle (CIC) model proposed by Mellman and Chen may serve as a useful guidepost [57]. This CIC comprises seven stepwise events, which are: release of cancer antigens from damaged or dying tumor cells (step 1); antigen presentation by dendritic cells (step 2); priming phase (T cell activation) (step 3); trafficking or migration of cytotoxic T lymphocytes (CTLs) to the tumor (step 4); infiltration of cytotoxic T lymphocytes into tumor tissue (step 5); recognition of cancer antigens presented by the HLA class I molecules of tumor cells (step 6); effector phase (destruction of tumor cells) (step 7). An eighth step is proposed in this review: reversal of immunosuppression, given how deleterious the effects of immunosuppression are on tumor-specific immune responses.

As shown in the figure below (Figure 2), *sine qua non* conditions for response to checkpoint inhibitors are the activation and enrichment of CTLs at the tumor sites, according to steps 1–5, indicative of a “hot” TME, as well as reversal or removal of immunosuppression (step 8).

Of all the treatment modalities that directly lyse tumor cells, such as chemotherapy, radiotherapy, targeted therapy, and oncolytic viruses, potentially leading to the release of tumor antigens from dying tumor cells for presentation to and activation of T cells (steps 1, 2, 3), only oncolytic viruses are self-amplifying. The importance of self-amplification is that it eliminates the need to infect every tumor cell at the treatment time since thousands and thousands of progeny infectious viruses emerge during cell lysis in a self-sustaining loop. This releases an abundance of pathogenic viral DNA and proteins as potential immune adjuvants to draw in responding immune cells (steps 4 and 5) as well as highly immunogenic tumor-specific antigens (TSAs) or neoantigens that can elicit T cell responses. Moreover, genome replication also amplifies therapeutic transgene protein production in situ [23].

Nevertheless, despite these immunomodulatory properties that suggest near-perfect complementarity with checkpoint inhibitors, oncolytic viruses have failed to completely deliver on their promise as immune sensitizers *par excellence*, which has engendered a degree of skepticism and disillusionment [58]. To the extent that oncolytic viruses (OVs) have disappointed/fallen short of expectations regarding therapeutic efficacy, it must be acknowledged that third and fourth generation OVs, armed with different therapeutic transgenes, represent a substantial improvement over first-generation versions, which lack them.

That said, OVs are a tractable platform, which lends itself to rational design and the advancement of strategies to address the following key challenges: (i) intense immunosuppression in and around the tumors, which “turns off” infiltrating T cells, (ii) “overengineering” of the oncolytic viruses for better safety and tumor selectivity, which potentially comes at the expense of potency (iii) the insertion into viruses of paradoxically dichotomous transgenes like GM-CSF, IL-12, IL-2, and TNF-a, which are ostensibly strong immune adjuvants but which also have been linked to immunosuppression, and (iv) downregulation or absence of the specific cell surface viral receptor on tumor cells, mediating viral entry, which in the case of adenovirus type V is CAR [59,60,61,62]. Added to these limitations is the rapid neutralization of seroprevalent viruses like adenoviruses by pre-existing antibodies and memory T cells when delivered systemically, although a reversible association with blood cells may protect adenoviruses, in particular, from opsonizing immunity [63]; also, minimal viral clearance has been demonstrated with intra-arterial administration [64].

A potential workaround to several of these challenges, since it is difficult, if not impossible, to design one virus which “does it all” in the absence of significant attenuation and loss of potency, is the sequential administration of different therapies that individually target specific steps of the cancer immunity cycle.

As an example, illustrated below in Figure 3, to maximize oncolysis and release of tumor antigen with priming and activation of T cells for steps 1 and 2, a more minimally modified oncolytic adenovirus such as AdAPT-001, whose replication kinetics out of all the clinical candidates that have been presented in this review are probably the most like wild type Ad5, might be administered initially. An additional potential benefit of AdAPT-001/AdAPT-039 is its neutralization of the immunosuppressive cytokine, TGF-β, which contributes to rampant T cell dysfunction in the TME (step 8).

Strong downregulation or absence of the coxsackie and adenovirus receptor (CAR) might instead prompt the use of non-CAR dependent viruses such as AdAPT-039, ONCOS-102, Enadenotucirev (EnAd), NG-350A, NG-641 or LoAd-703. As NG-641 expresses the chemokines CXCL9 and CXCL 10, which mediate immune cell trafficking [65], and LoAd-307 encodes the costimulatory ligands CD40L and 4-1BBL, the use of these viruses may also potentiate steps 4 (NG-641) and 6 and 7 (LoAd-307).

The administration of these OAVs would be followed approximately one week later, after peak viral load has been achieved, with the administration of an antiangiogenic therapy to restore normal vessel function for better immune effector cell infiltration in step 5. Since the so-called “vascular normalization window” is transient, lasting between 3–5 days with most antiangiogenic agents, checkpoint inhibitors might be administered during this time to enhance T cell recognition and cytotoxicity for steps 6 and 7.

Different permutations of this approximately 3-week cycle are possible. For example, separate oncolytic adenoviruses may work well together in a prime-boost regimen or combination with radiotherapy, chemotherapy, or targeted therapy. Furthermore, since the vascular normalization window may vary from patient to patient, the use of imaging technologies such as MRI (DCE-MRI and BOLD-MRI), dynamic contrast-enhanced ultrasonography (DCE-US), computed tomography, and positron emission tomography (PET) and serum markers such as soluble VEGFR (sFlt1) may determine it more precisely on an individual basis [66].

In conclusion, since 2011, when the first checkpoint inhibitor, ipilimumab (Yervoy), was approved, the oncology profession has been in the midst of a cold war with cancer since most tumors outside of melanoma and NSCLC are immunologically cold or checkpoint inhibitor unresponsive. The fundamental challenge in oncology is how, and using what combinatorial strategy, to transition from a cold war to a hot one so that the 70–80% of patients with tumors, which are currently checkpoint inhibitor unresponsive, benefit from them. In this context, oncolytic adenoviruses, which were introduced in 1996 and which target more than one of the steps of the cancer-immunity cycle, may prime the immunological landscape for more robust, long-lasting checkpoint inhibitor-led responses, especially in combination with canonical cytotoxics, targeted therapies, antiangiogenics, radiation therapy, and possibly other oncolytic viruses.

## Figures and Tables

**Figure 1 cancers-14-04701-f001:**
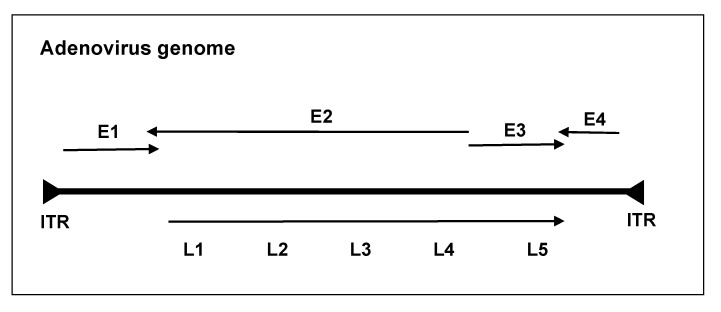
Simplified schematic of the human adenovirus genome. The adenoviral genome is linear and double-stranded and about 30–38 kb in length. Adenovirus genes are broadly organized into early and late transcription units based on their expression before or after DNA replication. The early transcription units include the early region, E1A, E1B, E2, E3, and E4, and late L1–L5. At each end of the genome are inverted terminal repeats (ITRs), which act as a primer for the host DNA polymerase.

**Figure 2 cancers-14-04701-f002:**
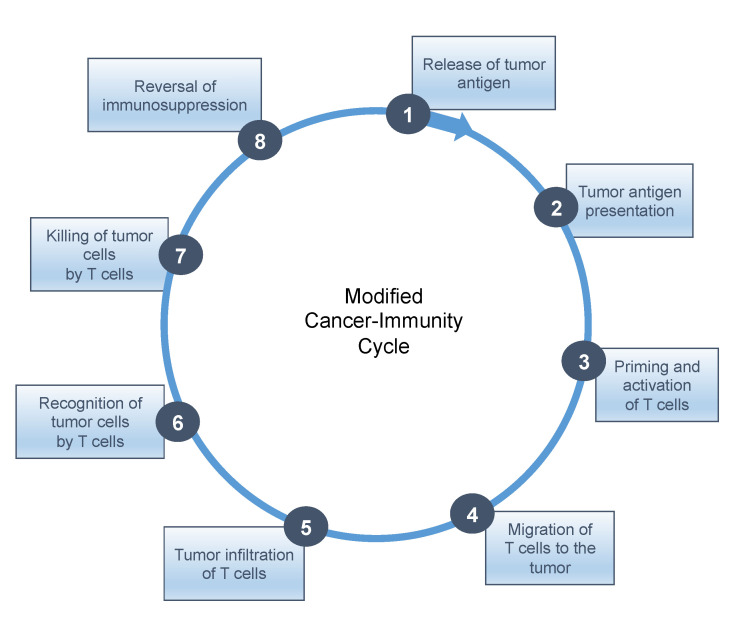
Modified cancer immunity cycle with the addition of immunosuppression.

**Figure 3 cancers-14-04701-f003:**
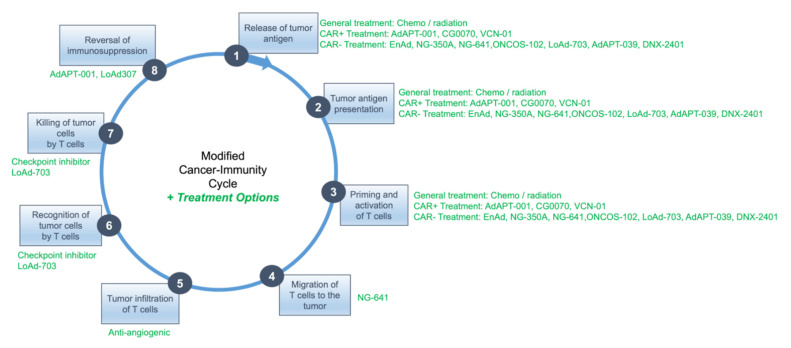
Therapeutic options based on cancer-immunity cycle.

**Table 1 cancers-14-04701-t001:** Examples of key design features and status of oncolytic adenoviral clinical candidates *.

Virus/ Indication	Company	Backbone	Tumor targeting	Promoter	Insertion Site/ Transgene(s)	Clinical Stage/ NCT Identifier	Combination with ICIs?
OAVs with clinical data							
AdAPT-001/ AdAPT-039/ TGF-β-driven solid tumors	EpicentRx	Ad5	50 bp deletion E1A	Native E1A	E1B19K/ TGF-β trap	Phase I/II/ NCT04673942	Yes
CG0070/ bladder cancer	CG Oncology	Ad5	24 bp deletion E1A	Exogenous E2F-1	E3/GM-CSF	Phase II/ NCT04452591 NCT04387461	Yes
Enadenotucirev/ recurrent platinum resistant ovarian cancer rectal cancer	Psioxus	Ad3	Ad 11 capsid	Native E1A	-	Phase I/ NCT02028117 NCT03916510 NCT02636036	Yes
NG-350A/ epithelial tumors	Psioxus	Ad3	Ad 11 capsid	Native E1A	E3-19K/ CD40 antibody	Phase I/ NCT03852511	Yes
NG-641/ epithelial tumors	Psioxus	Ad3	Ad 11 capsid	Native E1A	E3-19K/ FAP/CD3, CXCL9, CXCL10, IFNα	Phase I/ NCT04053283	Yes
ONCOS-102/ melanoma	Targovax	Ad5	Ad 3 capsid 24 bp deletion E1A	Native E1A	E1B19K/ GM-CSF	Phase I and Phase II/ NCT030036 NCT02963831	Yes
LoAd-703/ pancreatic cancer melanoma colorectal cancer	Lokon Pharma	Ad5	Ad 35 capsid	CMV	E3/ TMZ-CD40L, 4-1BBL	Phase I and Phase II/ NCT02705196 NCT04123470 NCT03555149	Yes
VCN-01	Synthetic Biologics	Ad5	24 bp deletion E1A RGD motif capsid	Not available	E3/hyalronidase	NCT02045589	No
OBP-301/ esophageal cancer esophagogastric adenocarcinoma	Oncolys BioPharma	Ad5	hTERT promoter	Exogenous hTERT	-	Phase I and Phase II/ NCT03213054 NCT03921021	Yes
DNX-2401/ CNS malignancies	DNAtrix	Ad5	24 bp deletion E1A RGD motif capsid	Unknown; not available	-	Phase I and Phase II/ NCT00805376 NCT03178032	Yes
CELYVIR ICOVIR-5 + MSCs/ pediatric solid tumors	Hospital Infantil Universitario Niño Jesús	Ad5	24 bp deletion E1A RGD motif capsid E2F-1 Mesenchymal stem cells (MSCs)	E2F-1	-	Phase I/ NCT01844661	No
Ad5-yCD/mutTKSR39rep-hIL12/ pancreatic cancer prostate cancer	Henry Ford Health System	Ad5	E1B-55K-deleted	Native	Mutant herpes simplex virus-thymidine kinase (HSV-tk), yeast cytosine deaminase (yCD), human interleukin-12 (hIL-12)	Phase I/ NCT03281382	No
**Virus/** **Indication**	**Company**	**Backbone**	**Tumor targeting**	**Promoter**	**Insertion Site/** **Transgene(s)**	**Clinical Stage/** **NCT Identifier**	**Combination with ICIs?**
**Other OAVs without clinical data**							
TILT-123/ melanoma solid tumors	TILT Biotherapeutics	Ad5	Ad3 fiber knob	Endogenous	E3/ TNFα-IRES- -IL-2	Phase I/ NCT04695327 NCT05271318 NCT05222932 NCT04217473	Yes
DNX-2440/ glioblastoma solid tumors	DNAtrix	Ad5	24 bp deletion E1A RGD motif capsid	Unknown	OX40	Phase I/ NCT04714983 NCT03714334 NCT02798406	No
CAdVEC/ HER2 positive tumors	Tessa Therapeutics	Ad5	24 bp deletion E1A	Unknown	-	Phase I/ NCT03740256	No; CAR-T cells
ORCA-10/ prostate cancer	Orca Therapeutics	Ad5	24 bp deletion E1A RGD motif capsid	Unknown	E3/19K-T1 protein	Phase I/II/ NCT04097002	No
SynOV1.1/ hepatocellular carcinoma	Beijing Syngentech Co.	Ad5	24 bp deletion E1A RGD motif capsid	AFP	GMCSF	Phase I/II/ NCT04612504	Yes

* Note that these are only examples chosen to highlight certain design features and so this table of OAVs is not all-encompassing. Abbreviations: Ad: adenovirus; bp: base pairs; IRES: internal ribosome entry site; TGF-beta: transforming growth factor beta; TNF-alpha: tumor necrosis factor-alpha; RGD: arginine–glycine–aspartic acid; AFP: alpha fetoprotein; HER2: human epidermal growth factor receptor 2; IL: interleukin; yCD: yeast cytosine deaminase; TK: thymidine kinase; MSC: mesenchymal stem cells; CMV: cytomegalovirus; GMCSF: granulocyte-macrophage colony-stimulating factor; CAR-T: chimeric antigen receptor-T; hTERT: human telomerase reverse transcriptase.

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
