# Peer review of "Oncolytic Adenoviruses: The Cold War against Cancer Finally Turns Hot"

_cancers, 2022, doi:10.3390/cancers14194701_

Round 1

Reviewer 1 Report

This review about oncolytic adenoviruses highlights their potential to turn unresponsive tumors into more responsive ones. The topic is of great interest for a broad audience and the manuscript is well written. However, I found a few minor concerns that authors should address:

1. Is the first sentence of the 'Introduction Section' necessary? Some people may find it inappropriate nowadays and, in my opinion, it is unnecessary in a research article.

2. All the figures should include a reference in the text.

3. Figure 1 should include the 'Ocolytic virus' symbol somewhere in the figure. One option could be on the arrow above the domino's boxes.

4. Figure 2 should be self-explanatory for a broad audience. I suggest to include in the legend part or all the information given in the text about the adenoviral replication cycle.

5. I found the 'Adenoviral Clinical Candidates' section very complex and too descriptive. If the authors could simplify it their work may reach a broader audience.

Author Response

Thank you for these comments, which are helpful.

1. Is the first sentence of the 'Introduction Section' necessary? Some people may find it inappropriate nowadays and, in my opinion, it is unnecessary in a research article. We have removed the first sentence

2. All the figures should include a reference in the text. Done

3. Figure 1 should include the 'Ocolytic virus' symbol somewhere in the figure. One option could be on the arrow above the domino's boxes. The oncolytic virus symbol was already in the figure. We wonder if the reviewer missed it? 

4. Figure 2 should be self-explanatory for a broad audience. I suggest to include in the legend part or all the information given in the text about the adenoviral replication cycle. This was a helpful suggestion and information was provided in the legend

5. I found the 'Adenoviral Clinical Candidates' section very complex and too descriptive. If the authors could simplify it their work may reach a broader audience. We were not sure how to do this specifically (since the comment is general) so we made no changes here. The point of this article was to describe the current state of play in oncolytic adenovirotherapy as well as to compare and contrast the different clinical adenoviral candidates, which is difficult to do if this section is simplified

Reviewer 2 Report

Tony Reid with colleagues from EpicentRx and clinical partners review the state-of-art of oncolytic adenoviruses (OAVs) in clinical development. Furthermore, they present a perspective on how engineered OAVs can be designed and utilized to strengthen anti-tumor immune activation by addressing mechanisms along the cancer immunity cycle paradigm.

The review provides an informative, focused insight into the present state of OAV clinical candidates and an expert perspective on key approaches for further development of OAVs toward clinical effectiveness as (combination) immunotherapies. Overall, the review is well written, focused, informative, timely and of high interest. However, some shortcomings should be addressed during revision:

-        the authors’ statements “adenoviruses have been honed to near perfection by eons of natural selection; this means the more adenoviruses are modified, the less well they infect and replicate” and “wild-type viruses are Darwinianly optimized for maximal infection, replication, and protein production; hence, any genomic and/or capsid modifications that attempt to improve on eons of evolutionary tinkering by “Mother Nature” likely reduce efficacy” are not correct and must be revised. There are several changes in the adenovirus genome that result in even dramatic increases in oncolytic activity (for example deletion of the E1B19K gene, overexpression of ADP, fiber chimerism…..). In this regard, it needs to be considered, for example, that adenoviruses have evolved to infect specific normal human tissues, not tumors. Studies have shown reduced adenovirus replication in tumor cells versus their normal host cells. Correspondingly, the scientific validity of the authors’ “Keep it simple and short, KISS” principle – best matched by the AdAPT-001 virus - is over-simplified and readers might get the impression biased, too, considering the authors’ conflict of interest.

-        The review (table 1) does not cover all present clinical studies exploring OAVs. The authors should make clear how they selected trials to be included in the review/table

-        The authors state that secondary viremia in their AdAPT-001 study suggests systemic delivery despite the presence of neutralizing antibodies. This is of course an indirect indication and only valid if viremia was determined as infectious particles rather than by PCR. Could the authors clarify this point?

-        He authors state that E3-deleted OAVs are significantly less active and refer to their KISS principle, but without referring to corresponding studies. The authors should provide references.

-        The description of EnAd as being HAdV-3 with HAdV-11p capsid is not correct

-        The description of ONCOS-102 should be revised: this virus contains a fiber shaft and tail domain of HAdV-5 and a fiber knob domain of HAdV-3. The GM-CSF gene replaces the E3 6.7K/gp19K gene.

Minor points:

-        The title does not well reflect the content, i.e. the overview of the clinical performance of lead AOVs.

-        The military language is certainly a matter of taste.

-        Abstract: “Intratumorally administered oncolytic viruses, colloquially referred to as “living drugs”, amplify…..”: This statement seems to be true for oncolytic viruses irrespective of the route of application.

-        Abstract: “….amplify themselves and, in some cases, the therapeutic gene that they carry….”: Are there cases, in which the therapeutic gene is not amplified?

-        Figure 1: the abbreviation “CDC” should be explained. Oncolytic virus appears in the legend, but not in the figure.

-        H101 is not mentioned in the paragraph on page 2 with viruses that have obtained marketing approval

-        Figure 2: the depiction of the transcription units seems to be over-simplified (E3)

-        The abbreviation OAV is not explained when used first and should be used consistently throughout the manuscript

-        Page 3: The modifications of E1A and E1B of conditionally replicative adenoviruses are not restricted to the promoters

-        Page 3 oncolytic adenoviruses cause mild ocular, respiratory or GI tract infections in immunocompetent individuals?

-        When stating viral doses as vps/ml, the actual administered dose is not clear

-        The statement that “….success stories are, in fact, relatively rare….” When referring to immune checkpoint inhibition does not seem to be adequate

-        Is there a rationale for positioning the addition of immunosuppression into the cancer immunity cycle at the chosen location?

Author Response

The authors would like to sincerely thank reviewer #2 for having clearly taken the time to read and analyze the manuscript in great detail and to offer constructive criticism about its shortcomings, about which, with one exception, related to the title, we agree. The reviewer's comments are in regular font, our responses are in italics.

  • the authors’ statements “adenoviruses have been honed to near perfection by eons of natural selection; this means the more adenoviruses are modified, the less well they infect and replicate” and “wild-type viruses are Darwinianly optimized for maximal infection, replication, and protein production; hence, any genomic and/or capsid modifications that attempt to improve on eons of evolutionary tinkering by “Mother Nature” likely reduce efficacy” are not correct and must be revised. There are several changes in the adenovirus genome that result in even dramatic increases in oncolytic activity (for example deletion of the E1B19K gene, overexpression of ADP, fiber chimerism…..). In this regard, it needs to be considered, for example, that adenoviruses have evolved to infect specific normal human tissues, not tumors. Studies have shown reduced adenovirus replication in tumor cells versus their normal host cells. Correspondingly, the scientific validity of the authors’ “Keep it simple and short, KISS” principle – best matched by the AdAPT-001 virus - is over-simplified and readers might get the impression biased, too, considering the authors’ conflict of interest. This point is well-taken and the specific language that the reviewer objected to, including references to KISS, has been deleted
  • The review (table 1) does not cover all present clinical studies exploring OAVs. The authors should make clear how they selected trials to be included in the review/table Another helpful comment. The title of the table has been modified with the word "examples" and asterisked with the phrase "Note that these are only examples chosen to highlight certain design features and so this table of OAVs is not all-encompassing"
  • The authors state that secondary viremia in their AdAPT-001 study suggests systemic delivery despite the presence of neutralizing antibodies. This is of course an indirect indication and only valid if viremia was determined as infectious particles rather than by PCR. Could the authors clarify this point? This is a common criticism. We have tried to measure viral titers in the blood but it has been too difficult. Viremia was inferred from PCR. Nevertheless, for clarity, the language about viremia was removed.
  • He authors state that E3-deleted OAVs are significantly less active and refer to their KISS principle, but without referring to corresponding studies. The authors should provide references. This sentence was removed
  • The description of EnAd as being HAdV-3 with HAdV-11p capsid is not correct. This description was removed
  • The description of ONCOS-102 should be revised: this virus contains a fiber shaft and tail domain of HAdV-5 and a fiber knob domain of HAdV-3. The GM-CSF gene replaces the E3 6.7K/gp19K gene. The reviewer's language was inserted word for word

Minor points:

  • The title does not well reflect the content, i.e. the overview of the clinical performance of lead AOVs. Understood but, respectfully, the title is meant to reflect the potential for AOVs to convert cold and immunologically unresponsive tumors into hot or immunologically responsive ones, which is a central theme of this manuscript that makes extensive use of the cancer immunity cycle. This cycle refers to a series of steps that need to occur for there to be antitumor immunity.
  • The military language is certainly a matter of taste. See above. The language is a play on words and intended to draw in readers and stimulate interest
  • Abstract: “Intratumorally administered oncolytic viruses, colloquially referred to as “living drugs”, amplify…..”: This statement seems to be true for oncolytic viruses irrespective of the route of application. Very true. The modifier "intratumorally administered" was deleted.
  • Abstract: “….amplify themselves and, in some cases, the therapeutic gene that they carry….”: Are there cases, in which the therapeutic gene is not amplified? This is a good catch. The phrase "in some cases" was deleted
  •  Figure 1: the abbreviation “CDC” should be explained. Oncolytic virus appears in the legend, but not in the figure. Abbreviations were added. Oncolytic virus appears in both the legend and the figure
  • H101 is not mentioned in the paragraph on page 2 with viruses that have obtained marketing approval. Added.
  • Figure 2: the depiction of the transcription units seems to be over-simplified (E3). This depiction simplified on purpose for readers that are not necessarily familiar with adenovirus and its genome
  • The abbreviation OAV is not explained when used first and should be used consistently throughout the manuscript. This is a good catch. Added.
  • Page 3: The modifications of E1A and E1B of conditionally replicative adenoviruses are not restricted to the promoters. The word "promoter" has been deleted.
  • Page 3 oncolytic adenoviruses cause mild ocular, respiratory or GI tract infections in immunocompetent individuals? Another good catch. This has been deleted.
  • When stating viral doses as vps/ml, the actual administered dose is not clear. mLs has been deleted.
  • The statement that “….success stories are, in fact, relatively rare….” When referring to immune checkpoint inhibition does not seem to be adequate. Understood and this phrase has been deleted.
  •  Is there a rationale for positioning the addition of immunosuppression into the cancer immunity cycle at the chosen location? Yes. The rationale is that the cancer immunity cycle of Chen and Mellman has been modified with the addition of immunosuppression so it made to add it in last. Also, reversal of immunosuppression is, in a way, the last (or one of the last) steps that needs to occur since even if T cells start to infiltrate the tumor it is potentially for nought if these infiltrating T cells are inactivated

Reviewer 3 Report

The authors present a large review on oncolytic adenovirus clinical candidates, their structures, mechanisms of action and their limitations in  terms of efficacy as monotherapy or in combination with various strategies.

First, the authors made a general introduction on clinically approved Oncolytic viruses and more precisely adenoviruses. They described their structure, advantages and limitations like the degree of immunosuppression of the tumor microenvironment in « cold » tumors.

Then, they described the different recombinant oncolytic adenovirus candidates worldwilde commercialized by different biotech and their phase I/II clinical trial safety profiles and benefits.

The authors have made an impressive work of bibliography summary discuting about the interest of combination in phase III trial with immune checkpoint modulators.

At the end, they present a nice figure higlighting the different therapeutical stategies options on cancer-immune cyle proposed by Melmann and Chen.

The review is original and can be published in the present form ;

However, just one comment, the authors should have mentionned in the text « figure 1, 2 etc..) and table 1.

Author Response

Thank you for this comment.

However, just one comment, the authors should have mentionned in the text « figure 1, 2 etc..) and table 1. This was added.